# Measuring Drug Therapy Effect on Osteoporotic Fracture Risk by Trabecular Bone Lacunarity: The LOTO Study

**DOI:** 10.3390/biomedicines11030781

**Published:** 2023-03-04

**Authors:** Annamaria Zaia, Pierluigi Maponi, Manuela Sallei, Roberta Galeazzi, Pietro Scendoni

**Affiliations:** 1Centre of Innovative Models and Technology for Ageing Care, Scientific Direction, IRCCS INRCA, 60121 Ancona, Italy; 2School of Science and Technology, University of Camerino, 62032 Camerino, Italy; 3Medical Imaging Division, Geriatric Hospital, IRCCS INRCA, 60121 Ancona, Italy; 4Analysis Laboratory, Geriatric Hospital, IRCCS INRCA, 60121 Ancona, Italy; 5Rheumatology Division, Geriatric Hospital, IRCCS INRCA, 63900 Fermo, Italy

**Keywords:** osteoporosis, fracture risk, osteoporosis treatment, vitamin D, drug therapy, bisphosphonates, trabecular bone microarchitecture, magnetic resonance imaging, fractal analysis, fractal lacunarity

## Abstract

An MRI method providing one parameter (TBLβ: trabecular-bone-lacunarity-parameter-β) that is sensitive to trabecular bone architecture (TBA) changes with aging and osteoporosis is under study as a new tool in the early diagnosis of bone fragility fracture. A cross-sectional and prospective observational study (LOTO: Lacunarity Of Trabecular bone in Osteoporosis) on over-50s women, at risk for bone fragility fracture, was designed to validate the method. From the baseline data, we observed that in women with prevalent vertebral fractures (VF+), TBA was differently characterized by TBLβ when osteoporosis treatment is considered. Here we verify the potential of TBLβ as an index of osteoporosis treatment efficacy. Untreated (*N* = 156) and treated (*N* = 123) women were considered to assess differences in TBLβ related to osteoporosis treatment. Prevalent VFs were found in 31% of subjects, 63% of which were under osteoporosis medications. The results show that TBLβ discriminates between VF+ and VF− patients (*p* = 0.004). This result is mostly stressed in untreated subjects. Treatment, drug therapy in particular (89% Bisphosphonates), significantly counteracts the difference between VF+ and VF− within and between groups: TBLβ values in treated patients are comparable to untreated VF− and statistically higher than untreated VF+ (*p* = 0.014) ones. These results highlight the potential role of TBLβ as an index of treatment efficacy.

## 1. Introduction

The World Health Organization (WHO) defines osteoporosis as a systemic skeleton disease characterized by bone mass loss and microarchitectural deterioration of bone tissue that induces bone fragility and increased susceptibility to fractures [1].

Bone fragility fractures account for increased morbidity and mortality in the geriatric population and perimenopausal women show the highest risk of early osteoporosis onset [2,3,4]. In fact, postmenopausal osteoporosis is peculiar to 50–65-year-old women because of the fast resorption of trabecular bone, which is associated with estrogen deficiency. The pattern of bone fragility fracture in postmenopausal osteoporosis primarily involves vertebras, whereas senile osteoporosis, affecting over 65-year-old subjects, is mainly characterized by fractures at the hip and femur, when deterioration of bone tissue also involves the cortical bone [5].

It should be pointed out that bone deterioration is a silent process and osteoporosis is often diagnosed after a painful bone fracture occurs [4]. It is known that the first fracture event worsens the risk of new fractures and speeds up their onset. In addition, vertebral fractures (VFs) are generally painful, accompanied by a loss of function, and can even happen in the absence of severe symptoms. In any case, VFs often recur, and the number of fractures affects the consequent increase in disability [6].

In clinical practice, areal bone mineral density (BMD) estimated using dual-energy X-ray absorptiometry (DXA) represents the first choice for osteoporosis diagnosis. Nevertheless, there is evidence that BMD alone is not effective for fracture risk prediction as the incidence of osteoporotic fractures does not always correlate with low BMD [7]. Moreover, osteoporosis drug treatment to prevent or reduce bone loss does not always decrease bone fracture risk [8]. As a matter of fact, bone quality factors, such as bone tissue composition and microstructure, are responsible for osteoporotic fracture risk. In particular, trabecular bone microarchitecture (TBA) has emerged as an important contributor to bone fragility independently of BMD; therefore, TBA studies could help to understand the mechanisms of bone deterioration and the action of drugs in preventing bone fragility fractures [9].

High-resolution magnetic resonance imaging (MRI) represents a noninvasive-nonionizing tool to characterize TBA. Most studies on TBA characterization in osteoporosis using MRI mainly deal with the prevalence and incidence of bone fragility fractures and TBA changes induced by osteoporosis therapy. From pioneering studies on MRI characterization of bone structure, parameters of TBA allowed separating fractured and non-fractured osteoporotic patients better than BMD [10,11,12]. The most critical endpoint in measuring the effect of osteoporosis treatment is the incidence of bone fragility fractures as it needs years to be reached. In addition, MRI methods proposed for TBA characterization mainly were based on classic histomorphometric texture analyses that produce a large number of parameters that are difficult to analyze [11,13]. The hard interpretation of such a set of parameters has been limiting the diffusion of a promising noninvasive-nonionizing tool in clinical practice. Due to the difficulty in investigating TBA degeneration, a more affordable index to predict osteoporotic fracture risk is needed and BMD has become a surrogate index for this type of assessment. Nevertheless, there is evidence for BMD limitations, while TBA characterization using MRI better describes changes induced by antiresorptive therapy [14,15,16,17]. 

An MRI method developed in our institute provides one parameter, namely TBLβ (trabecular bone lacunarity parameter *β*), able to detect TBA deterioration induced by aging and osteoporosis [18,19,20,21]. The diagnostic validation of the method as a new tool for early diagnosis of osteoporotic fractures is in progress [22]. The TBLβ method, measuring fractal lacunarity of TBA in MRI spin-echo images of vertebras, has emerged as a fast and easy noninvasive-nonionizing promising tool for assessing osteoporotic fracture risk and is potentially useful for monitoring treatment efficacy [21,23].

The LOTO (Lacunarity Of Trabecular bone in Osteoporosis) study is a cross-sectional and prospective observational study designed for the diagnostic validation of the TBLβ method. The TBLβ results from the baseline data showed that the contribution of TBA degeneration to prevalent vertebral fractures in over-50s women was statistically higher than BMD [22]. In particular, we observed that patients with prior VFs showing TBLβ values > 40, that are at low risk for fracture, included several treated patients. Therefore, in the present study, we focused on untreated (T−) and treated (T+) LOTO subjects to verify the potential of TBLβ in assessing the efficacy of osteoporosis therapies on bone fragility fracture risk.

## 2. Materials and Methods

### 2.1. Participants and Procedures

Over-50s osteopenic/osteoporotic women, known to be at risk for bone fragility fracture, with/without osteoporosis treatment were considered among patients recruited for the LOTO study. The LOTO study is a cross-sectional and prospective observational study designed for the diagnostic validation of the TBLβ method based on fractal lacunarity analysis of TBA in MRI spin-echo axial sections of lumbar vertebra images, as a new tool potentially useful for the early diagnosis of bone fragility fracture risk. 

The study design and procedures, approved by the Ethics Committee of our Institute (FiORdiLOTO SC/11/281), have been previously widely described [22]. Briefly, over-50s women with a BMD-DXA diagnosis of osteopenia/osteoporosis were recruited according to the inclusion/exclusion criteria as follows: inclusion criteria were age ≥ 50 years; BMD T-score between −1 and −2.5 (osteopenia) or T-score equal to −2.5 or lower (osteoporosis); primitive osteoporosis, with or without prevalent vertebra fragility fractures; and written informed consent provided. Exclusion criteria regarded: osteoporosis secondary to drug-induced bone loss, chronic diseases or genetic diseases; MRI contraindications; and severe cognitive and/or functional impairment.

Demographic and clinical data were recorded by interview during the first visit. Then, patients underwent the regular diagnostic practice for osteoporosis and osteoporotic fracture: spine and femur BMD assessment using DXA; blood analysis to exclude secondary osteoporosis and for therapy monitoring; and dorsal-lumbar spine X-ray morphometry for VF diagnosis according to Genant’s criteria [24]. After inclusion in the study, patients underwent an MRI scan of the L1-L4 spine to acquire vertebra images for TBA characterization.

### 2.2. Characterization of Trabecular Bone Microarchitecture

MRI scans of the dorsal-lumbar spine were performed using high-resolution MRI, 1.5T whole body system (Gyroscan Intera; Philips-Medical System, ACR-Nema 1.0) using a phased array dS Spine coil. Axial section images of vertebral bodies were acquired using the spin-echo multislice technique (about 10 slices with a thickness of 3 mm without a space gap between slices) to visualize TBA. The pulse sequence was TE = 15 ms, TR = 525 ms; flip angle = 90°, matrix = 512 × 512, and pixel size = 0.469 mm for a scan time shorter than 15 min. 

MRI spin-echo images were stored in the shared informatics folder along with other medical imaging and clinical material to be processed and analyzed. TBLβ, our index of fracture risk, was computed on lumbar vertebra images by fitting gliding box curvilinear plot *Λ(b)* [25] with our bio-mathematical model based on hyperbola model function:(1)$$L(b;\alpha ,\beta ,\gamma )=\frac{\beta }{b^{\alpha }}+\gamma ,\;\;\;\;b\in [b_{\min},b_{\max}]$$
where (*α, β, γ*) are the fitting parameters that characterize any structure analyzed. In particular, parameter *β*, our TBLβ, describes the concavity of the curve and quantifies lacunarity. It is worth noting that high-*β* values are obtained for low lacunarity, that is, low-fracture risk; on the other hand, low-*β* values mean high lacunarity, that is, high-fracture risk (for details see Zaia et al. [18,19]). The TBLβ method was calibrated by using the grayscale version on the middle axial section(s) of the fourth lumbar vertebra (L4) [21,22].

### 2.3. Main Endpoint Outcome

The TBLβ of our bio-mathematical model was the main outcome measure as an index of TBA degeneration to assess osteoporosis treatment efficacy on fracture risk. It was calculated as the average of results from the two central L4 axial sections (5th and 6th out of 10) by means of the grayscale version of our method [21,22] in patients with at least one treatment (Vitamin D and/or calcium supplements, VitD/Ca, or drug therapy with/without VitD/Ca) and compared to untreated subjects.

### 2.4. Statistical Analyses

The sample size of the LOTO study, to assess the diagnostic capacity and accuracy of the TBLβ method, was estimated at 280 osteopenic/osteoporotic patients, with a 20% prevalence of VFs in over-50s women, a non-relevant VF incidence during a two-year follow-up, and a 10% drop out. The sample size was calculated based on 0.05 first-type error and a study power higher than 80%. For more details see Zaia et al. [22].

SPSS package (v. 19, SPSS Inc. Chicago, IL, USA) was used for statistical analyses and a statistical significance level *p* ≤ 0.05 was considered. Demographic and clinical characteristics of the whole patient sample were summarized by usual descriptive statistics.

Univariate analysis was performed for continuous variables. Differences between groups were compared using the Student’s *t*-test and Chi-squared test. The nonparametric alternative Mann–Whitney U test was applied when normal distribution, checked by the Kolmogorov–Smirnov test, was not acceptable.

The best cut-off value of TBLβ to predict VFs was defined as equal to 40 based on the Youden index in the ROC curve and median value from the whole sample [22] and used to separate patients with high-/low-fracture risk.

## 3. Results

### 3.1. Demographic and Clinical Characteristics

All subjects eligible for the LOTO study were included in this study. Table 1 summarizes the main demographic and clinical characteristics of the whole sample and of the two main groups considered: VF+, with prevalent vertebral fracture (*n* 88, 31.5%) and VF−, without vertebral fractures (*n* 191, 68.5%).

The age range of the whole sample was 50–85 years (mean age ± SD equal to 60 ± 7). A similar age distribution was observed in VF− (range 50–85 years, mean age 59 ± 7 years) while VF+ (age range 51–80 years) showed a statistically higher mean age (63 ± 7 years, *p* = 0.0003) than VF−.

Lumbar spine BMD T-score was found equal to or lower than −2.5 (osteoporosis) in 48.1% of women. VF+ patients accounted for 45.8% of subjects defined as osteopenic (T-score > −2.5) at the lumbar spine by DXA-BMD and 65.1% younger than 65 years.

Dealing with osteoporosis treatment, 123 out of 279 patients (44.1%) were with at least one osteoporotic medication, 35.8% of which with VitD/Ca alone and, within the drug therapy group, 88.6% with bisphosphonates treatment. Other medications accounted for 11.4%. Among the overall treated subjects (T+), 57% were VF− (mean age 61 ± 7 years; range 51–85 years) and 43% were VF+ (63 ± 8 years; range 51–80 years). The distribution of VF− and VF+ within each treatment group is detailed in Table 2.

### 3.2. TBLβ as an Index of Osteoporosis Therapy Efficacy

The TBLβ results, summarized in Table 3, show that the proposed method was able to discriminate between VF+ and VF− patients (*p* = 0.004). This result was confirmed in untreated T− subjects (*p* = 0.027). 

Treatment, any medication (T+), and drug therapy in particular, significantly counteracted the difference between VF+ and VF− within and between subgroups. These results are shown in Table 3 and Figure 1. The TBLβ values of treated VF+ were comparable to untreated VF− patients (*p* = 0.48) and were statistically higher than untreated VF+ (*p* = 0.014).

It is worth noting that the positive effect of treatment (any medication) on TBA quality, as measured by our index TBLβ, is mainly in charge of drug therapy. In fact, VitD/Ca did not affect TBA as TBLβ values in VF+ patients treated with VitD/Ca alone were statistically lower than VF− ones within (*p* = 0.005) and between (p = 0.002) groups and even statistically lower than untreated VF+ subjects (*p* = 0.034).

Interestingly, preliminary 1-year prospective results show that, among T− patients at baseline that started to receive treatment (76 out of 86), 22% had at least one incident VF+, 71% of which were associated with a baseline at risk TBLβ value (≤40). These results further stress the goodness of TBLβ as an index of fracture risk.

## 4. Discussion

In this study, we present the potential role of TBLβ as an index that is useful for monitoring osteoporosis treatment efficacy in reducing bone fragility fracture risk. The TBLβmethod has been recently presented as a new diagnostic tool that is useful for assessing osteoporotic fracture risk (LOTO study) [22]. The method is based on the fractal lacunarity analysis of TBA in lumbar vertebra axial images acquired using the 1.5T-MRI spin-echo technique. It produces one parameter, namely TBLβ, particularly sensitive to TBA degeneration with both aging and osteoporosis [19,20,21].

The baseline results from the LOTO study, a cross-sectional and prospective observational study on osteopenic/osteoporotic over-50s women, designed for the diagnostic assessment of the method, show that TBLβ separates prevalent VF+ and VF− subjects better than BMD. Furthermore, preliminary 1-year prospective results suggest that TBLβ is able to predict incident VF better than BMD [22]. In addition, we observed that TBLβ values differ between T− subjects and the whole sample comprising T+ patients.

This aspect prompted us to look further into the potential role of TBLβ as an index for monitoring osteoporosis treatment efficacy on bone fragility fracture risk by comparing T− and T+ subgroups. Results from this study show that TBLβ, as an index of TBA degeneration and predictor of bone fragility fracture for values ≤40, can also represent a valid index for treatment assessment. In fact, any medication (T+), and drug therapy in particular, significantly counteracts the difference between VF+ and VF− within and between subgroups with TBLβ values comparable to untreated VF− patients and statistically higher than untreated VF+ ones. It is worth noting that the positive effect of osteoporosis medications on TBA deterioration is mainly due to drug therapy, mostly represented by bisphosphonates, whereas VitD/Ca alone are ineffective if not deleterious on TBA as shown by TBLβ values that are even statistically lower than untreated VF+ patients.

Results from this study are consistent with the literature in the field and further stress the potential of TBLβ as an index useful in the assessment of therapy efficacy. As a matter of fact, guidelines for osteoporosis drug interventions, being cost-effective on a population basis [26,27], are well defined: drug therapy can be considered if the woman has a previous fragility fracture, a BMD DXA T-score ≤ −2.5, or T-scores between −1 and −2.5 in the presence of high-fracture risk [28]. At present, bisphosphonates represent the first-line option of drug therapy in osteoporotic patients [29] and are the preferred intervention in over-60s women [30]. The antiresorptive action through the mechanism of inhibiting osteoclast-mediated bone resorption makes bisphosphonates adequate for osteoporosis treatment [31] and several randomized trials show that bisphosphonates reduce the risk of fractures [32].

Dealing with VitD/Ca literature, vitamin D supplement represents one of the most common therapeutic interventions to reduce falls and fractures. In spite of numerous studies in the field, inconclusive results support the reduction in fractures induced by vitamin D supplementation [33]. In particular, vitamin D (especially vitamin D3) alone can reduce the incidence of falls but not fracture risk. Fracture risk reduction can be only induced by vitamin D when administered in association with calcium [34]. Nevertheless, from a meta-analysis of randomized clinical trials by Zhao et al. [35], it emerges that supplements of vitamin D or calcium alone or in combination, compared with placebo or no treatment, do not reduce the risk of fractures in community-dwelling older adults. The authors concluded that the lack of a significant association between calcium, vitamin D, or combined supplements and the incidence of hip, vertebral, nonvertebral, or total fractures does not further support the routine use of vitamin D and/or calcium supplements in community-dwelling older adults [35].

Our results on VitD/Ca are consistent with the lack of benefit in fracture risk reduction induced by such a treatment. In fact, TBLβ, our index of TBA integrity and predictive of fracture risk, in patients treated with VitD/Ca alone, does not significantly differ from untreated subjects. Nevertheless, before the decision is made to exclude the routine use of vitamin D and/or calcium supplements [35], we have to take into account that bone belongs to the skeletal-muscle apparatus and bone injuries unbalance the integrity of the apparatus itself where muscle plays its role. Increasing evidence supports vitamin D action just on muscle [36]; therefore, vitamin D supplements can still represent a useful integrative treatment mainly in the case of osteo-sarcopenia. In this context, it has to be highlighted that the research on new osteoporosis drug therapies has been devoted to new molecules acting on both bone and muscle, such as romosozumab [37,38].

The screening of the population at risk for bone fragility and treatment assessment to prevent fractures are important tools to both improve life quality in the elderly and lighten the related healthcare-socio-economic burden. Noninvasive tools are necessary to characterize bone quality. It would allow the accurate monitoring of the individual’s risk of bone fragility fracture by evaluating osteoporosis progression and treatment efficacy [8,39,40]. The WHO recommended DXA-BMD measurement for osteoporosis diagnosis before noninvasive technologies for in vivo assessment of bone structure were available. In spite of recently updated guidelines for osteoporosis management by also contemplating bone quality [39,41,42], BMD and age are still the primary risk factors considered for bone fragility fracture. Dedicated algorithms and calculators can be adopted to overcome the limits of these two risk factors and better predict fracture risk as well as to decide on osteoporosis treatment [43,44]. For instance, the FRAX algorithm is a common tool for fracture risk assessment that estimates the 10-year probability of hip and major osteoporotic fractures on the basis of the individual’s risk factor profile [43]. Nevertheless, there is evidence that FRAX does not improve fracture risk assessment when compared to BMD in peri- and early postmenopausal women [45,46]. Last-generation DXA devices have been equipped with software dedicated to bone quality assessment by TBS (trabecular bone score). However, this technology, despite its name, cannot investigate TBA and, in particular, it has been emerging that it fails in assessing the effect of osteoporosis drug therapies [47,48].

Quantitative characterization of TBA by nonionizing-noninvasive tools in clinical practice would complement DXA-BMD methods and complete the diagnosis of osteoporosis as defined by the WHO [1] by assessing and monitoring longitudinal changes. Several features of MRI candidate this technology as a noninvasive-nonionizing tool for in vivo study of human bone tissue: it does not use ionizing radiation, allows direct acquisition of multiplanar images, and can investigate bone physiology otherwise not explorable by other imaging techniques [11]. Several different MRI methods have been proposed to analyze bone tissue in osteoporosis [49,50,51], often characterized by numerous parameters to be calculated and analyzed. In particular, classic morphometric parameters, such as bone volume fraction (BV/TV), trabecular bone number (Tb.N), and trabecular thickness (Tb.Th), resulted in being more effective than BMD in discriminating groups with/without fractures [11,12]. The lack of longitudinal studies with large cohorts further inhibits the transition of these methods into clinical practice because of the rare evidence for their usefulness in fracture risk prediction. As a matter of fact, a most recent prospective study on alendronate treatment used 3.0 T MRI for image acquisition of mirror sites (distal tibia, distal radius, and proximal femur). The TBA parameters analyzed included BV/TV, Tb.N, Tb.Th, and Tb.Sp (trabecular spacing) and seven parameters by geodesic topological analysis (GTA). Only apparent Tb.N and four GTA parameters in the distal tibia were statistically affected by treatment after 24 months when compared to BMD [52].

Dealing with TBA characterization, the proposed TBLβ method represents a very promising tool. It uses 1.5 T MRI, which is widely available in clinical settings and provides one parameter particularly sensitive to TBA changes, thus representing a suitable tool for an easy and fast transfer into both research and clinical fields. The limits and advantages of the method have already been widely discussed elsewhere [21,22]. It is notable that image processing and image analysis techniques allow for overcoming the limits of image quality and resolution. In particular, the computational approach adopted in the TBLβ method to quantify TBA deterioration, based on fractal lacunarity texture analysis in grayscale images, overcomes the limits of the image binarization process and provides one parametric result, TBLβ, a holistic estimate of TBA, comprehensive of BV/TV, Tb.Sp, Tb.N, Tb.Th. In fact, lacunarity, a term from the Latin *lacuna* (lack or hole), coined by Mandelbrot [53] to describe gap distribution in a fractal, by measuring the space-filling capacity of a complex object, can describe bone network discontinuity as well as the sizes of trabecular bone marrow spaces [19,54], the changes of which are an index of bone fragility fracture.

In this study, TBLβ also emerges as a useful index to assess the action of osteoporosis treatment on TBA deterioration. Therefore, the TBLβ method has the potential for monitoring the efficacy of osteoporosis therapy administered to prevent or reduce fracture risk. It is worth noting, however, that these baseline results are from an observational study and, hence, are characterized by heterogeneity in both therapy type and time. Therefore, further studies are needed to confirm the role of TBLβ as an index of therapy efficacy.

## 5. Conclusions

In this study, we further stress the goodness of TBLβ as an index of bone fragility fracture risk useful in the assessment of therapy efficacy [22,55] and highlight the potential role TBLβ can play in monitoring the effect of osteoporosis drug therapy in preventing or reducing bone fragility fracture risk. In fact, antiresorptive drug therapy, mainly represented by bisphosphonates, can counteract TBA deterioration as measured using TBLβ that shows comparable values in patients with/without prior VFs. Vitamin D and/or calcium supplements alone, instead, fail in recovering/counteracting TBA deterioration that characterizes subjects with prevalent VFs.

More consistent results on TBLβ as an index of therapy efficacy can be expected from the prospective LOTO study; in fact, information on osteoporosis therapy prescribed to untreated patients will be accurately documented. It would allow establishing the usefulness of TBLβ for therapy monitoring by assessing TBA changes and fracture incidence in single individuals to evaluate the efficacy of the treatment administered and, therefore, the appropriateness of therapeutic prescription.

In any case, pharmacological trials designed on purpose are required. In this context, the TBLβ method, based on fractal lacunarity texture analysis of MRI-TBA, is easy and fast to apply, thus facilitating these kinds of studies.

## Figures and Tables

**Figure 1 biomedicines-11-00781-f001:**
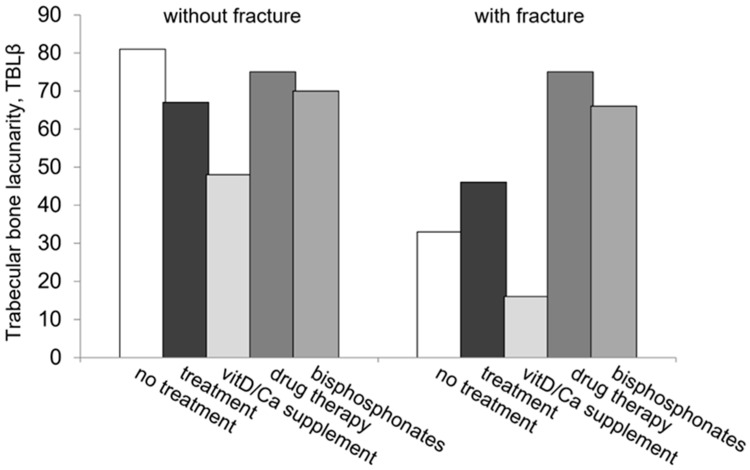
The effect of osteoporosis treatment on TBA as measured using TBLβ in over-50s women with (right) and without (left) prevalent bone fragility fracture. TBA: trabecular bone microarchitecture; TBLβ: trabecular bone lacunarity parameter *β* as an index of fracture risk.

**Table 1 biomedicines-11-00781-t001:** Baseline demographic and clinical characteristics of LOTO patients.

Characteristic	Overall *N* = 279	VF− *N* = 191	VF+ *N* = 88
**Age, years**			
Mean ± SD	60 ± 7	59 ± 7	63 ± 7
<65, *n* (%)	212 (76.0)	153 (80.1)	59 (67.0)
≥65, *n* (%)	67 (24.0)	38 (19.9)	29 (33.0)
**Bone mineral density, T-score**			
Lumbar spine L1-L4			
Mean ± SD	−2.4 ± 0.9	−2.4 ± 0.9	−2.5 ± 0.8
>−2.5, *n* (%)	133 (47.7)	87 (45.5)	46 (52.3)
≤−2.5, *n* (%)	146 (52.3)	104 (54.5)	42 (47.7)
**Trabecular bone architecture, TBLβ**			
Lumbar spine L4			
Median (interquartile range)	43 (26–87)	51 (27–103)	31 (23–60)
>40, *n* (%)	143 (51.3)	113 (59.2)	30 (34.1)
≤40, *n* (%)	136 (48.7)	78 (40.8)	58 (65.9)
**Osteoporosis medications**			
none, *n* (%)	156 (55.9)	121 (63.4)	35 (39.8)
any medication, *n* (%)	123 (44.1)	70 (36.6)	53 (60.2)

VF−, VF+: without, with prevalent vertebral fragility fracture; TBLβ: trabecular bone lacunarity parameter *β*; interquartile range: Q1–Q3.

**Table 2 biomedicines-11-00781-t002:** Osteoporosis medications.

Treatment	Overall *N* = 123		VF− *N* = 70		VF+ *N* = 53	
Vitamin D/calcium supplements, *n* (%) ^a^	95	(77.2)	55	(78.6)	40	(75.5)
Vitamin D/calcium supplements alone, *n* (%) ^a^	44	(35.8)	25	(35.7)	19	(35.8)
Drug therapy, *n* (%) ^a^	79	(64.2)	45	(64.3)	34	(64.1)
Bisphosphonates, *n* (%) ^b^	70	(88.6)	43	(95.6)	27	(79.4)
alendronate, *n* (%) ^c^	24	(34.3)	16	(37.2)	8	(29.6)
clodronate, *n* (%) ^c^	20	(28.6)	9	(20.9)	11	(40.7)
ibandronate, *n* (%) ^c^	6	(8.6)	4	(9.3)	2	(7.4)
neridronate, *n* (%) ^c^	3	(4.3)	2	(4.6)	1	(3.7)
risedronate, *n* (%) ^c^	17	(24.3)	12	(27.9)	5	(18.5)
Other, *n* (%) ^b^	9	(11.4)	2	(4.4)	7	(20.6)
strontium ranelate, *n* (%) ^c^	7	(77.8)	1	(50.0)	6	(85.7)

VF−, VF+: without, with prevalent vertebral fragility fracture; ^a^: % within treated patients; ^b^: % within drug therapy group; ^c^: % within related drug therapy subgroup.

**Table 3 biomedicines-11-00781-t003:** TBLβ as an index of therapy efficacy on osteoporotic fracture risk.

Subjects			TBLβ		
	*n* VF−/VF+ (%)		VF−	VF+	*p*
Overall	191/88	(100)	51 (27–103)	31 (23–60)	0.004
Untreated	121/35	(55.9)	52 (28–110)	30 (23–56)	0.027
Treated (any medication)	70/53	(44.1)	50 (26–93)	31 (23–69)	0.08
Vit D/calcium supplements alone ^a^	25/19	(35.8)	45 (18–66)	27 (23–39)	0.005
Drug therapy ^a^	45/34	(73.2)	52 (26–101)	37 (25–101)	0.31
Bisphosphonates ^b^	43/27	(88.6)	50 (26–100)	39 (24–125)	0.40
Alendronate ^c^	16/8	(34.3)	50 (28–93)	41 (19–152)	**0.98**

TBLβ: trabecular bone lacunarity parameter *β*; VF−, VF+: without, with prevalent vertebral fracture; ^a^: % within treated patients; ^b^: % within drug therapy group; ^c^: % within related drug therapy subgroup; TBLβ values are median (interquartile range Q1–Q3) *p*: statistical significance for *p* ≤ 0.05 from Mann–Whitney U test.

## Data Availability

The datasets generated and analyzed during the current study are available from the corresponding author upon reasonable request.

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
