# Peer review of "Measuring Drug Therapy Effect on Osteoporotic Fracture Risk by Trabecular Bone Lacunarity: The LOTO Study"

_biomedicines, 2023, doi:10.3390/biomedicines11030781_

Round 1
Reviewer 1 Report
well-structured scientific work with valid applied research and a not indifferent scientific purpose. the scientific community has always wanted to research suitable and appropriate therapeutic plans for the treatment of pathologies, therefore in this case being able to have a diagnostic tool to support not only the clinic but also and above all the therapy and prognosis is fundamental. the reference to an ethics committee is fundamental and precise, the methods of inclusion and exclusion of patients are well described, the diagnostic evaluation methods for measuring the structure and bone composition are excellently described, finally the statistical analyzes performed using the SPSS package are excellent and well described by researchers. the results obtained were excellent and of considerable interest to the international scientific community. In fact, this study underlined the goodness of the diagnostic-evaluation method as an index of bone fragility and fracture risk, useful in assessing the efficacy of the therapy as well as the role it can play in monitoring the effect of osteoporosis drug therapy to prevent or reduce the risk of fragility fractures.
Author Response
Please, see the attachment

Reviewer 2 Report
Authors have investigated a new tool in the early diagnosis of bone fragility fractures using a cross-sectional and prospective observational study.
Overall, the paper is well written. I have a few comments.
1) The sample size determination: Authors should address how the sample size was determined. What is statistical power?
2) Introduction is a bit lengthy. Just briefly discuss the context and the summary of the literature related to this topic. Perhaps some of the information presented in the introduction could be moved to the discussion
3) Tables are stand-alone and self-sustaining. That means a reader should understand the data presented in the table without referring to the manuscript's text. So, please add more footnotes at the bottom of the table with appropriate superscripts embedded in the text of the table. Also, the table footnote should contain the type of statistical test used, abbreviations used, whether the data were mean ± SD or SE, and the significance level.
4) Table 3 is crowded a bit. No need for IQR. Just give Q1 and Q3. Q1 should be listed first and Q3 later.
5) Limitations of this study should be mentioned in the discussion at the end.
6) Abstract and text of the manuscript: Purpose, methods, results, and literature (except for the well-known facts) should be stated in the past tense.
7) I feel there are too many acronyms used. Please reduce these acronyms if possible. It takes so much time to comprehend these various acronyms. It is a bit distractive.
8) For non-significant p-values, just use 2 decimals. For significant values, use up to 3 decimals.
Author Response
Please, see the attachment

Reviewer 3 Report
In the Introduction, combine lines 33-48 into one paragraph.
There are some minor grammatical errors in the manuscript, such as line 59 “in the last decades”, line 87 “show”, line 88 “is”, line 90 “is”, and line 102. Check the full manuscript and correct the errors.
Author Response
Please, see the attachment

Round 2
Reviewer 2 Report
I have reviewed the revised paper. I am satisfied with the revisions, although tables readability could be improved a bit.